# Advantages and Challenges of Using AI Planning in Cloud Migration

**Hongtan Sun, Maja Vukovic, John Rofrano, Chen Lin**
IBM T.J. Watson Research Center
1101 Kitchawan Rd,
Yorktown Heights, New York 10598
hongtan.sun@ibm.com, { maja, rofrano } @us.ibm.com, liana.lin@ibm.com

## Abstract

Cloud Migration transforms customer's data, application and services from original IT platform to one or more cloud environment, with the goal of improving the performance of the IT system while reducing the IT management cost. The enterprise level Cloud Migration projects are generally complex, involves dynamically planning and replanning various types of transformations for up to 10k endpoints. Currently the planning and replanning in Cloud Migration are generally done manually or semi-manually with heavy dependency on the migration expert's domain knowledge, which takes days to even weeks for each round of planning or replanning. As a result, automated planning engine that is capable of generating high quality migration plan in a short time is particularly desirable for the migration industry. In this short paper, we briefly introduce the advantages of using AI planning in Cloud Migration, a preliminary prototype, as well as the challenges the requires attention from the planning and scheduling society.

## Introduction

Automated planning and AI planning have been investigated extensively by researchers and successfully applied in many areas for decades, for example, health care (Cardoen, Demeulemeester, and Beliën 2010), semiconductor manufacturing (Uzsoy, Lee and Martin-Vega 1992), and aviation (Bazargan, M. 2010), to name a few. Meanwhile, attracted by the promise of the scalability, flexibility and potentially lower cost of the resources, more and more enterprises are considering moving their IT infrastructure and applications to Cloud or Hybrid Cloud service platforms, which is called *Cloud Migration* in general (Armbrust et. al. 2010, Khajeh-Hosseini, Greenwood, and Sommerville 2010). Noticing that the discussions of using AI planning in the Cloud Migration are limited both in academia and in industry, in this short paper we identify the advantages and challenges of applying AI planning to Cloud Migration by (i) introducing Cloud Migration and its planning problem; (ii) demonstrate problem feasibility by showing a prototype AI planning model; and (iii) discuss the limits of current model and future research.

---

## Planning in Cloud Migration

Cloud Migration transforms customer's data, application and services from original IT platform, hosted on servers hosted in-house or cloud environment, to one or more cloud environment, with the goal of improving the performance of the IT system while reducing the IT management cost. Generally speaking, enterprise-level Cloud Migration is a complex and usually long running process that requires careful planning.

Cloud Migration includes four major steps: Discovery, Planning, Execute and Validation (Vukovic and Hwang 2016). In the *Discovery* stage, migration experts investigate the current IT system, collect data and identify customer's migration goals. Then migration experts allocate resources and schedule the executions activities, refer to as the *Planning* stage. Next the migration are executed as planned, which is called *Execute* stage. In the last *Valication* stage, all the applications are tested that they are running as expected on the new cloud environment. Due to the complexity of the IT system and IT infrastructure, there may not be clear boundaries between the major steps. It could happen that in the Planning stage some data inconsistency were observed and additional discoveries are performed and the migration execution needed to be re-scheduled or re-planned.

Due to the complexity of migration projects, low tolerance on errors and its heavy dependence on the migration expert's domain knowledge, current practitioners mostly perform migration planning either manually. or using tools manually created runbooks (Transition Manager, Velostrata). For example, in Transition Manager, the user has to upload manually scripts, e.g. groovy scripts (The Apache Groovy Programming Language), and ask the tool to generate runbook for existing wave bundles. Meanwhile, Velostrata Manager would create a .csv format template for the user to manually put tasks in and create the runbook (Creating and modifying runbooks). These planning and replanning approaches rely heavily on the practitioner's previous migration experience and domain knowledge, hence is not scalable.

With the fast evolution of computing speed and machine learning technologies, domain-independent AI planners are becoming more and more powerful (Ghallab, Nau,

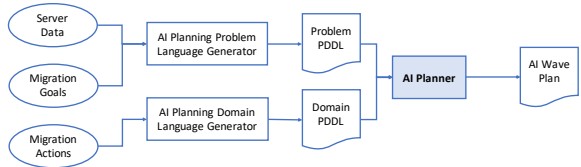

Figure 1: Overview for the prototype AI planner

and Traverso 2004). Better planners emerge and demonstrate their performance and capability every year on the International Planning Competition (ICP) and the International Conference on Automated Planning and Scheduling (ICAPS). As result, taking advantage of the domain-independent planners to automate the planning of Cloud Migration is extremely desirable for migration practitioners.

## Prototype AI Planner

In a Cloud Migration planning problem, there are $N$ assets to be migrated. Assets may communicate with each other, for example, an application reads/writes a database, hence causes dependencies between assets and enforces precedence constraints for migration tasks. For instance, if asset $A$ depends on asset $B$, the migration of asset $A$ has to be done before asset $B$'s migration. The goal of migration planning is to allocate resources and create sequence of tasks to be executed. In the case of enterprise level migration, the execution should be performed in a limited time window to minimize potential business disruption.

From application point of view, the main step in developing an AI planner based on domain-independent planner is to formulate the planning problem in Cloud Migration as a planning problem for the planner. In one of our prototype AI planner, a simplest scenario, in which only migration of physical servers and virtual machines are considered, is modeled as Domain file and Problem file using Planning Domain Definition Language (PDDL). The objects in the domain file are server and wave. Each server has is assigned a numeric value called 'effort hours', which represents the cost of migrating this server. Each wave is assigned a numeric value called 'effort hour limit', which enforces a capacity constraint for the number of servers to be migrated in each wave. The goal is to migrate all the servers without violating the capacity constraint.

A planner supported by Metric-FF planner is developed to test the performance and a graphical UI is created for users to upload spreadsheet containing server's information (Jackson, Rofrano, Hwang, and Vukovic 2018). In particular, translation engines are developed to generate the Domain.pddl file and Problem.pddl file automatically. When there are limited number of servers, the planner finds solution in a few seconds. However, when tested with 500 servers, the planner did not find any solution in 2 hours. Figure 1 shows an overview of the prototype planner.

## Challenges and Future Research Directions

In conclusion, in this short paper, the Cloud Migration process is investigated and a prospective research direction is identified around using domain-independent AI planner in Cloud Migration Planning. Automate migration planning is desirable for practitioners from both the cost perspective and the quality consideration. It also brings in new research topics. Some of them are listed as following.

- Optimize the modeling of migration planning problem so that the domain and problem file can be generated faster, shorten the auto-planning time.

- Noticing that many top-performed planners in ICP does not support metric feature, efficient algorithms that removes the metric requirements in the resource planning part of a migration planning problem needs to be developed.

- Improve planner's computational speed or develop algorithms so it can generate migration plan for thousands assets and more complicated migration scenarios.

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
