# OpenReview forum: "Advantages and Challenges of Using AI Planning in Cloud Migration"
_icaps-conference.org/ICAPS/2019/Workshop/SPARK — SPARK 2019_

### Official Review · AnonReviewer1 · 2019-04-25
**A new application domain for AI Planning**

**Rating:** 4
**Confidence:** 2

**Review:**

In this paper the authors provide an overview of challenges and opportunities of exploiting AI planning for supporting cloud migration. Cloud migration is the process of moving data, applications and services from an original IT platform to a cloud environment. The process usually involves 4 main steps, that are briefly described in the paper.
The authors suggest that AI Planning could be used in the (surprise surprise) planning step of the cloud migration process. They outline a basic framework that could be used for this purpose and highlight future challenges.

The area of application is particularly interesting, and I am not aware of previous planning use in the domain. While the short paper is (necessarily) at a very high level, I'm confident the paper could provide a valuable contribution to the workshop, and can foster some interesting discussions.

---

### Official Review · AnonReviewer2 · 2019-04-30
**More details please!**

**Rating:** 3
**Confidence:** 2

**Review:**

This paper proposes a (PDDL) planning model of the problem of migrating IT services into a cloud environment.

The paper is well written over all, but it is lacking details about the planning model. It is only 2 pages long. While I appreciate short papers, I think that in this case it would be preferable to use another page or two to provide a detailed explanation of the actions and planning decisions that are encoded in the PDDL planning model, and perhaps a small example of a migration problem and the planning dilemmas it poses.  It would also be good if the authors make their PDDL model (domain and example problems) publicly available: both because it gives the planning community a further application benchmark problem, and also because it provides a complete understanding of how the problem is formulated.

If the paper is accepted for the workshop, I hope the authors will adopt at least one, preferably both, of these suggestions to provide readers with the full details of their planning problem and its formulation.

---

### Official Review · AnonReviewer3 · 2019-04-30
**potentially interesting topic; submission is light on details**

**Rating:** 3
**Confidence:** 2

**Review:**

This short submission (2 pages) seeks to motivate the use of AI planning technology to support the migration of computing resources to a Cloud environment. It outlines the problem at a high level, briefly describes an initial prototype built on Metric-FF for a simple version of the problem, and summarizes open challenges determined from the development of that first prototype.

Cloud Migration could potentially be a good application for AI planning but the case isn't made in a completely convincing manner in this paper. One issue is that the reader isn't presented with a clear description of what a planning model would need to encompass.  The problem is characterized as "allocate ... resources ... and the migration expert's work efforts so all the assets can be migrated ... and running seamlessly as they were before the migration".  Only sequencing constraints are mentioned as being essential to the model but presumably there are also capacity/throughput constraints?  And isn't optimization important as well (cost, speed)?  Recognizing that this is a short paper, it still should be possible to provide more clarity on the nature of the planning constraints that are inherent to the problem.

PROs:
•	Potentially interesting application domain to present to the community
•	The authors have explored the space sufficiently that they should be able to share some insights into the challenges and opportunities

CONs:
•	Submission is light on content, in particular it lacks sufficient detail about what defines the core planning/scheduling problem

Some minor points:

- It is mentioned that current practice is to plan manually or using tools called "runbooks" - it would be helpful to know more about what those tools do.

 - The criteria used for ordering the References is unclear - it is neither alphabetical, chronological, nor reference order.

---

### Decision · Program_Chairs · 2019-05-08
**Acceptance Decision**

**Decision:**

Accept

**Comment:**

Reviewers have suggestions to improve the work and the presentation, but still treats a potential application.